# Features of Structure and Absorption in the Jet-Launching Region of M87

**Wei Zhao** [1,2,*] **, Xiaoyu Hong** [1,2,3,4] **, Tao An** [1,2] **, Xiaofeng Li** [1,2,3,4] **, Xiaopeng Cheng** [1,2,3] **and Fang Wu** [1,2]

1   Shanghai Astronomical Observatory, 80 Nandan Road, Shanghai 200030, China; xhong@shao.ac.cn (X.H.); antao@shao.ac.cn (T.A.); lixf@shao.ac.cn (X.L.); xcheng@shao.ac.cn (X.C.); wufang@shao.ac.cn (F.W.)
2   Key Laboratory of Radio Astronomy, Chinese Academy of Sciences, Nanjing 210008, China
3   University of Chinese Academy of Sciences, 19A Yuquanlu, Beijing 100049, China
4   ShanghaiTech University, 100 Haike Road, Pudong, Shanghai 201210, China
*   Correspondence: weizhao@shao.ac.cn

**Abstract:** M87 is one of the best available source for studying the AGN jet-launching region. To enrich our knowledge of this region, with quasi-simultaneous observations using VLBA at 22, 43 and 86 GHz, we capture the images of the radio jet in M87 on a scale within several thousand $R_s$. Based on the images, we analyze the transverse jet structure and obtain the most accurate spectral-index maps of the jet in M87 so far, then for the first time, we compare the results of the two analyses and find a spatial association between the jet collimations and the local enhancement of the density of external medium in the jet-launching region. We also find the external medium is not uniform, and greatly contributes to the free-free absorption in this region. In addition, we find for the jet in M87, its temporal morphology in the launching region may be largely affected by the local, short-lived kink instability growing in itself.

**Keywords:** galaxies: individual (M87); galaxies: jets; galaxies: active; radio continuum: galaxies

## 1. Introduction

M87 is a nearby (D∼16.7 Mpc, [1]) FRI radio galaxy which harbors a central black hole with a mass reported from 2.4 to $7.2 \times 10^9$ M$_{sun}$ [2–4]. Profiting from its proximity, the angular resolution of ground-based Very Long Baseline Interferometry (VLBI) now is down to a few times of the angular size of schwarzschild radii ($R_s$) of the super massive black hole (SMBH) in M87. In recent years, more and more facts of the vicinity of the SMBH in M87 are unveiled by high angular-resolution VLBI observations especially at millimeter wavelength. e.g., the accurate position of SMBH in M87 is constrained by [5]; the central compact radio source in M87 is imaged directly by [6].

M87 is a prototype to study the AGN jet-launching region as well as the vicinity of the SMBH. The structure of the jet in its launching region has been determined with VLBI monitoring, that smears out the variability details by stacking multi-epoch observations at 22, 43 and 86 GHz [7–9]: the limb-brightened jet structure starts at a projected distance down to $7R_s$ to the SMBH, with an apparent opening angle wider than 100° [9,10]; as the limb-brightened jet propagates outward, it has to experience multiple expansions followed with subsequent recollimations [8,9] i.e., "collimation regimes" and finally reaches an equilibrium parabolic expansion in several thousand $R_s$, and keeps the shape until $10^5 R_s$ [11,12]. Besides the motion along the jet, the jet flow rotates around the axis clockwisely, and the toroidal component of magnetic field

is found close to the core by VLBI polarimetry [8]. Additionally, the counterjet is proved to be existent and with a limb-brightened structure as well [8].

In this work, we present images of the radio jet in M87 on a scale within 10 mas, i.e., a few thousand $R_s$, captured by very long baseline array (VLBA) at 22, 43 and 86 GHz quasi-simultaneously. Based on these images with high dynamical ranges, we analyze the transverse jet structure and obtain the most accurate spectral-index maps of the jet in M87 so far, and for the first time, we compare the results of the two analyses and find spatial associations between the structural features and the absorption features. We then make discussions on these associations. We also discuss the cause of the temporal jet morphology at the observed epoch. Throughout this work, we use cosmological parameters $H_0 = 67.8$ km s$^{-1}$ Mpc$^{-1}$, $\Omega_M = 0.308$, $\Omega_\Lambda = 0.692$, then 1 pc corresponds to 11.3 mas, and $R_s \approx 7 \times 10^{-4}$ pc $= 7.9$ μas if we take $M_{BH} = 7.2 \times 10^9$ M$_{sun}$.

## 2. Observations and Data Reduction

In 28 March 2015, we observed M87 at 22 GHz ($\lambda = 1.3$ cm) and 43 GHz ($\lambda = 7$ mm) with VLBA. To optimize the $u - v$ coverage, the two observing frequencies were cycled in each of the observing runs. Six days later, we observed M87 at 86 GHz ($\lambda = 3$ mm) with VLBA. For all three frequencies, some antennas were absent during the observations due to some technical problems. The details of the problems are described in the footnote of Table 1.

**Table 1.** Observations.

| Date | Frequency (GHz) | Telescopes | On-Source Time (Min) | Beam Size [a] (mas × mas, deg) | $I_{peak}$ [b] (mJy beam$^{-1}$) | $\sigma$ [c] (mJy beam$^{-1}$) |
|---|---|---|---|---|---|---|
| 28 March 2015 | 22.72 | VLBA [d] | 124 | 0.93 × 0.45, 12.6 | 1054 | 0.360 |
| | 43.12 | VLBA [d] | 124 | 0.56 × 0.23, 16.7 | 726 | 0.224 |
| 3 April 2015 | 86.28 | VLBA [e] | 289 | 0.24 × 0.12, −17.7 | 501.4 | 0.217 |

Notes: [a] The Synthesized beam FWHM size of the images. [b] The peak flux density on the images. [c] The root-mean-square noise of the images. [d] Nine antennas were used, MK was absent due to the "bad diskpack" problem. [e] Seven antennas were used, HN and SC have no 86 GHz receiver while NL was "disc failure".

An automatic reference pointing control was used for the antenna-pointing. 5-min scans on the nearby bright source 3C 273 and 3C279 as fringe finders, delay and bandpass calibrators, were inserted during the observations, every 35 min for 22/43 GHz and 75 min for 86 GHz. The received signals were sampled with 2-bit quantization and recorded with an aggregate data rate of 2048 Mbit s$^{-1}$. See details of observations in Table 1.

The initial data calibration was made in AIPS (Astronomical Image Processing System) software package. For 22 and 43 GHz, we followed the standard procedure of VLBA data reduction. For 86 GHz, the procedure is slightly different: the global fringe-fitting was first performed on scans of 3C273 and 3C279 with point source models to derive time evolutions of the residual delay, rate, and phase for each IF separately; the derived residual delay varies slowly with time, so the solutions of 3C 273 and 3C279's delay were used as a first-order approximation for M87's residual delay; then the global fringe-fitting was performed twice on M87, the first was performed with a point source model, the solution interval was set to 30 s and the threshold of signal to noise ratio (SNR) was set to 3.0, to avoid false signals, a tight delay and rate search window (10 nanoseconds and 20 mHz for delay and rate respectively) were used, the IFs are averaged to increase the SNR by a factor of 2.8; then we produce an initial image, and perform the second one with the CLEAN models obtained from the initial map to increase the fringe detection rate (the detection rate was increased slightly for 0.2%).

After calibration, multi-source data are split into single-source files and imported into DIFMAP for self-calibration and imaging. Self-calibration/Imaging loops are performed for multiple iterations to obtain images with high dynamical ranges. The final image was produced in DIFMAP with natural weighting.

## 3. Results

We show the resulting VLBI images in Figures 1 and 2 as contour plots overlapping with pseudo-color images of the same frequency respectively. We list the parameters of the images (e.g., synthesized beam size, peak flux density and root-mean-square noise $\sigma$) in Table 1. Some structural features in the jet are apparently seen, e.g., on the 22 GHz image, the jet shrinks locally between 1–2 mas from the center of the image, and the northern limb is interrupted between 5 and 7 mas from the center of the image; on the 43 GHz image, the jet bends southward between 1–2 mas from the center of the image; on the 86 GHz image, the northern limb looks rather ragged, while the shape of its southern counterpart is far better defined; emission at the northeast of the most luminous region which is probably the hint of the counter jet, is seen on images at all three frequencies .

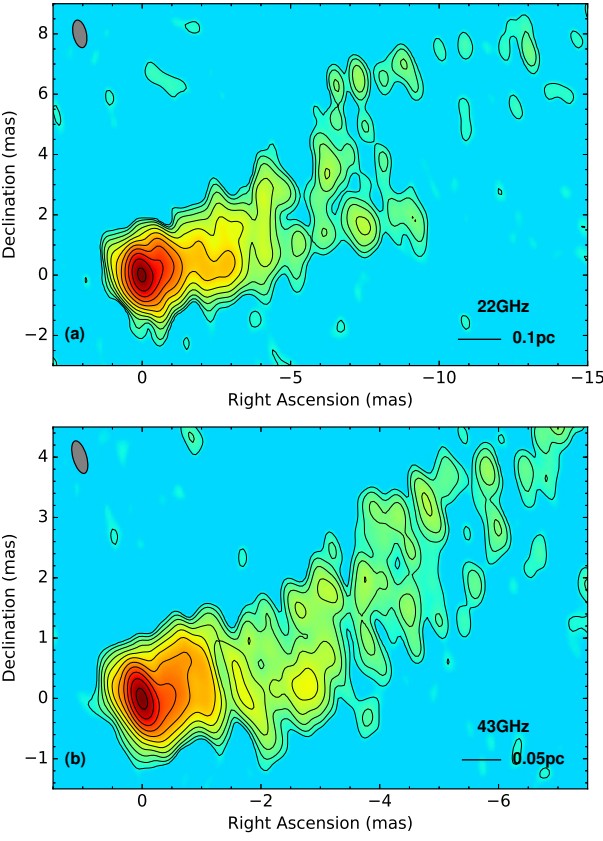

**Figure 1.** 22 and 43 GHz contour plots overlapping with pseudo-color images of the same frequency respectively, with the synthesized beam shown at the top-left corner of each panel. Panel (**a**) shows the image of M87 at 22 GHz with a synthesized beam of $0.93 \times 0.45$ mas at a position angle of $12.6°$; contours in this image are from $(-1, 1, 2, 4, 8...) \times 0.86$ mJy beam$^{-1}$. Panel (**b**) shows the image of M87 at 43 GHz with a synthesized beam of $0.56 \times 0.23$ mas at a position angle of $16.7°$; contours in this image are from $(-1, 1, 2, 4, 8...) \times 0.54$ mJy beam$^{-1}$.

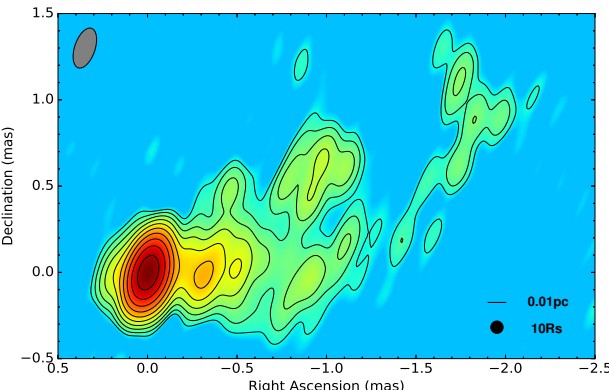

**Figure 2.** 86 GHz contour plot overlapping with pseudo-color image of the same frequency of M87; contours are from $(-1, 1, 2, 4, 8...) \times 0.60$ mJy beam$^{-1}$. The synthesized beam of $0.24 \times 0.12$ mas at a position angle of $-17.7°$ is shown at the top-left corner.

For the further analysis, we fit the most luminous region on the images as an elliptical Gaussian with procedure MODELFIT in DIFMAP, and refer it as "core" hereafter. We list the fitting results of the parameters in Table 2 with their uncertainty estimated as in [13]. We also estimate the brightness temperature of the core as in [14]. The brightness temperature of the core is estimated to be at the magnitude of $10^{10}$ K for all three frequencies. This is the same as given in [9,10], which is nearly one order of magnitude lower than the equipartition brightness temperature.

**Table 2.** Core Component.

| Frequency (GHz) | Flux Density $^a$ (mJy) | Radius $^b$ (μas) | Theta $^c$ (deg) | Major Axis $^d$ (μas) | Axial Ratio $^e$ | Phi $^f$ (deg) | Brightness Temperature $^g$ ($10^{10}$K) |
|---|---|---|---|---|---|---|---|
| 22 | $1605 \pm 195$ | $19.9 \pm 2.1$ | $-80.8 \pm 6.1$ | $479.2 \pm 1.1$ | 0.84 | 43.4 | 3.1 |
| 43 | $1016 \pm 122$ | $3.6 \pm 0.4$ | $-142.8 \pm 5.8$ | $261.2 \pm 0.2$ | 0.62 | 20.2 | 2.3 |
| 86 | $580 \pm 58$ | $3.6 \pm 0.3$ | $-44.7 \pm 4.4$ | $73.0 \pm 0.2$ | 0.52 | 31.8 | 5.0 |

Notes: $^a$ Integral flux density of Gaussians. $^b$ Distance of Gaussians to the center of the images. $^c$ Position angle of Gaussians referring to the north. $^d$ Size of Major axis of Gaussians. $^e$ Axial ratio of Gaussians. $^f$ Position angle of major axis of Gaussians referring to the north. $^g$ Brightness temperature of Gaussians.

### 3.1. Transverse Jet Structure

In this part, we use intensity slices to extract details of the jet and to analyze the transverse jet structure evolving with core distance $d$. To improve the angular resolution in the direction transverse to the jet with the least distortion of the images, in advance of slicing, we restored the images with circular Gaussian beams whose diameter equal to the geometric mean of the length of major and minor axis of their synthesized beams, which is 0.65, 0.36 and 0.17 mas for the image at 22, 43 and 86 GHz respectively. The restored images are shown as contours in the left panels of Figure 3. Then we slice the restored images in a direction perpendicular to the approximate overall jet axis, which is $-67°$ respect to the north (referred as "jet axis" hereafter). For the 22 GHz image, we made slices between $d = 2.00$ and 10.00 mas, at an interval of 0.05 mas; for the 43 GHz image, we made slices between $d = 0.10$ and 3.00 mas, at an interval of 0.05 mas; for the 86 GHz image, we made slices between $d = 0.10$ and 1.40 mas, at an interval of 0.01 mas. The right panels of Figure 3 present the sample slices, showing the intensity as a function of position angles referring to the jet axis (PAs, the northern limb is at positive values, while its southern counterpart is at negative values). Locations of these sample slices are presented as lines superimposed on the contours in the left panels of Figure 3.

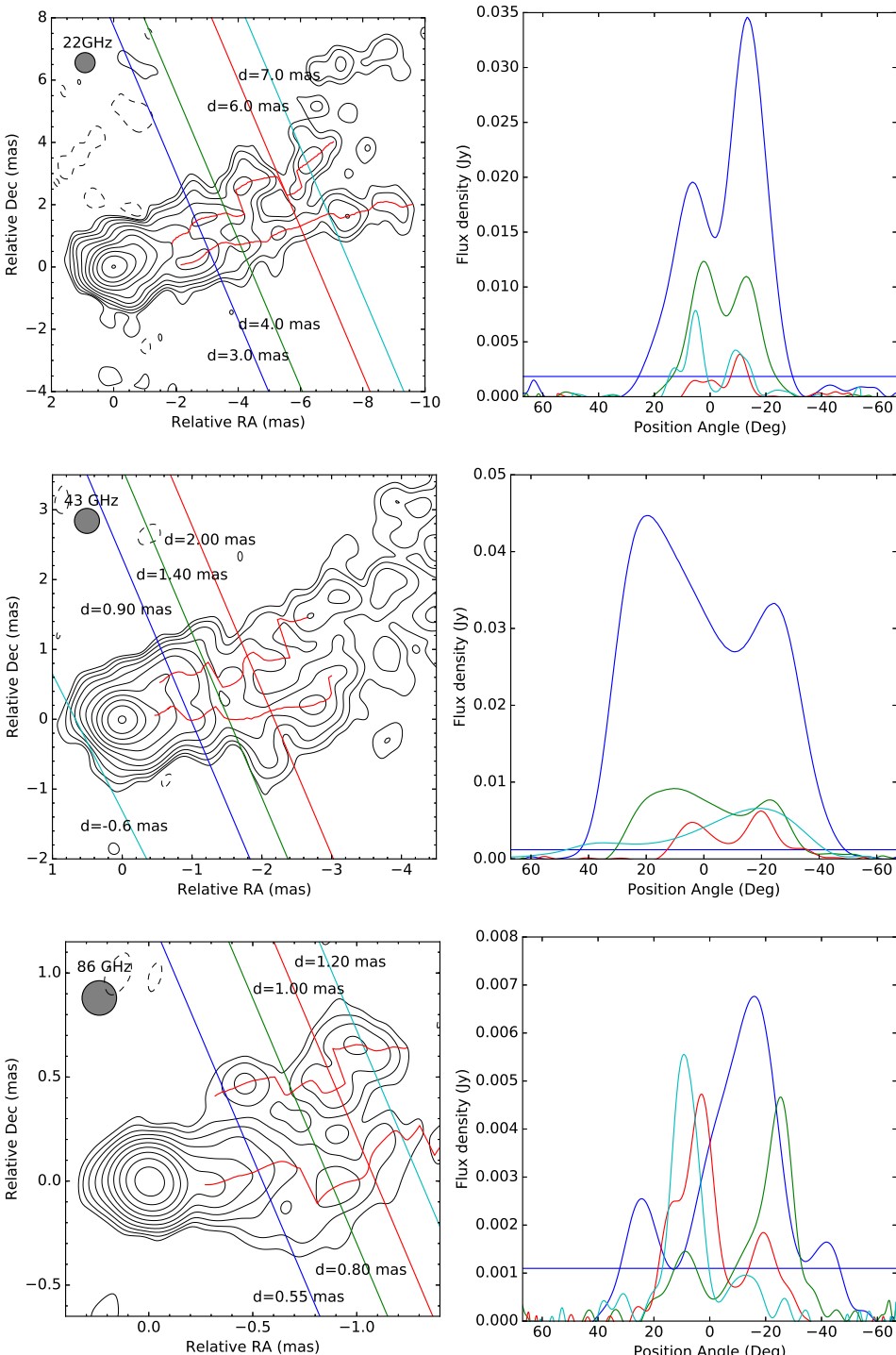

**Figure 3.** The left panels shows the images restored with circular beams as contours. The restored beams are shown at the top-left corner of each panel. Locations of sample slices are presented as lines superimposed on the contours. The ridgeline is presented as red curves superimposed on the contour plots. The right panels present sample slices, showing the intensity as functions of position angles PAs referring to the jet axis. The northern limb is at positive values, while its southern counterpart is at negative values. The blue horizontal lines represent $5\sigma$ of the restored images.

In our intensity slices, at 43 and 86 GHz, the northern limb represents as a well defined hump since $d = 0.65$ and 0.50 mas respectively, while its southern counterpart since $d = 0.45$ and 0.24 mas respectively. The counter jet emission also seems bifurcated on the restored 43 GHz image, and a slice at $d = 0.60$ mas shows this bifurcated structure clearly (see the cyan line in the second row of Figure 3). We also recover the ridgeline of the limb-brightened jet in M87 with the intensity slices. The ridgeline is presented in the left panels of Figure 3 as red curves superimposed on the contour plots. As our result shows, both limbs are tortuous within $d = 1.30$ mas; as going outward, the northern limb is still tortuous, while its southern counterpart becomes smooth.

We present PAs of the intensity peak of both limbs as a function of $d$ in Figure 4. The general trend is both limbs approach to the jet axis gradually as going outward, but there are some local structures seen on both limbs. We also notice that, if the structure measured at higher frequency is moved outward for 0.1 or 0.2 mas, then it looks quite consistent with that measured at the adjacent lower frequency. This apparent shift could possibly be attributed to a resolution effects which occur if there are gradients in the intensity for a jet. This is a combination of gradients in intensity, coupled with angular resolution depending on frequency. Within $d = 1.30$ mas, at 86 GHz, the northern limb is deflected northward at $d = 1.05$ mas, while at 43 GHz, the corresponding deflection is seen at $d = 1.30$ mas; at 86 GHz, the southern limb veers southward at $d = 0.62$ and 1.18 mas respectively, while at 43 GHz the corresponding changes are seen at $d = 0.75$ and 1.30 mas respectively. Beyond $d = 1.30$ mas, the northern limb becomes fragmentized, and its orientation changes abruptly at least at three locations ($d = 1.45, 2.55$ and $4.55$ mas), a wiggle is seen between $d = 1.70$ and 2.30 mas, and an interruption is seen around $d = 6.00$ mas; on the contrary, the southern limb evolves smoothly as going outward, only a mild wiggle is seen between $d = 3.00$ and 4.00 mas, and its PA tends to be stable at $\sim -10°$ since $d = 4.00$ mas.

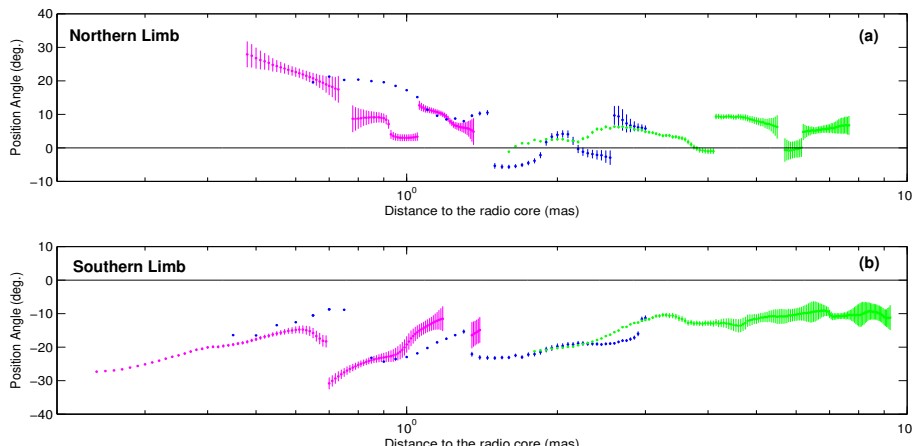

**Figure 4.** (**a**) Position angles of the northern limb in M87 as functions of core distance $d$. (**b**) Position angles of the southern limb in M87 as functions of core distance $d$. The colors represent the measured frequencies, magenta for 86 GHz, blue for 43 GHz and green for 22 GHz. The uncertainty of PA is estimated as $\frac{20°}{\text{SNR}}$, in which $20°$ is a typical width of the jet limb, and SNR is the ratio between the intensity peak and the $\sigma$ of images. only peaks with intensity above $5\sigma$ are presented to make sure the result is reliable.

In the following analysis, we move the jet structure measured at 86 GHz outward for 0.15 mas, and the structure measured at 22 GHz inward for 0.25 mas to eliminate the effect brought by different angular resolutions. We define the apparent jet opening angle $\psi_{app}$ as the difference of PAs of two limbs. The panel (a) of Figure 5 shows $\psi_{app}$ as a function of $d$. The parabolic collimation profile, $\psi_{app} \propto r^{0.58}$, determined

by [11,12], is also presented for reference. The overall jet generally evolves around the parabolic profile, but some local structural features are clearly seen. So we divide the jet into six segments, and each segment includes an expansion and a following recollimation: segment 1 includes an quick expansion between $d = 0.75$ and 0.90 mas, followed by a gradual recollimation between $d = 0.90$ and 1.30 mas; segment 2 includes an expansion between $d = 1.30$ and 1.45 mas, followed by a quick recollimation between $d = 1.45$ and 1.50 mas; segment 3 includes an gradual expansion between $d = 1.50$ and 2.00 mas, followed by a gradual recollimation between $d = 2.00$ and 2.55 mas; segment 4 includes a quick expansion between $d = 2.55$ and 2.60 mas, followed by a slow recollimation between $d = 2.60$ and 3.35 mas; segment 5 includes an expansion between $d = 3.35$ and 3.65 mas, followed by a gradual recollimation between $d = 3.65$ and 4.25 mas; segment 6 includes an expansion between $d = 4.25$ and 4.60 mas, followed by a recollimation beyond $d = 4.60$ mas. We define the apparent offset of jet center $\delta_{app}$ as the arithmetic average of the PAs of two jet limbs as well. The panel (b) of Figure 5 shows $\delta_{app}$ as a function of $d$, and the jet axis is presented as a black solid line for reference. As the Figure shows, the jet bends southward since $d = 0.75$ mas, the offset reaches its maximum at $d = 1.45$ mas (where it is most clearly seen on the 43 GHz image), where the northern limb veers south, and gradually returns to zero around $d = 3.00$ mas.

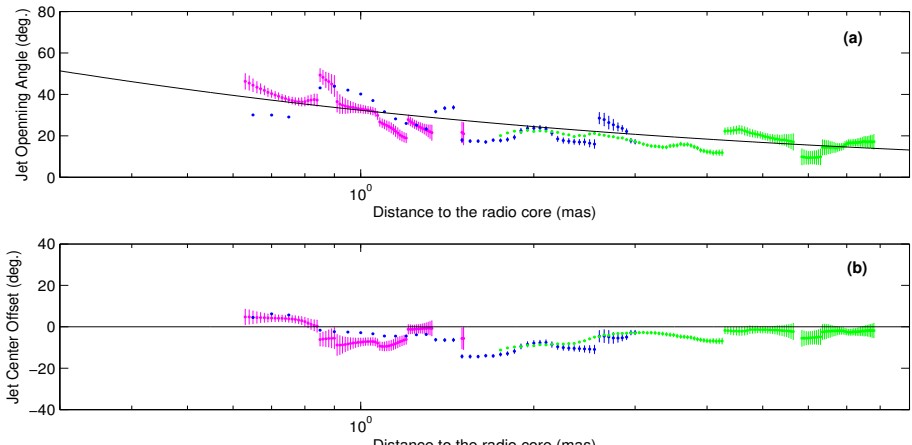

**Figure 5.** (**a**) The apparent opening angle $\psi_{app}$ as functions of core distance $d$. Parabolic collimation profile determined by Asada & Nakamura is shown for reference. (**b**) The jet center offset $\delta_{app}$ as functions of core distance $d$. The jet axis is are shown for reference. The colors represent the measured frequencies: magenta for 86 GHz, blue for 43 GHz and green for 22 GHz. The uncertainties of $\psi_{app}$ and $\delta_{app}$ are estimated as $\sqrt{\Delta PA_{North}^2 + \Delta PA_{South}^2}$. Only peaks with intensity above $5\sigma$ are presented to make sure the result is reliable.

By comparing the behaviors of each limbs with $\psi_{app}$ evolving with $d$, we find that, structural features including the expansion and recollimation in segment 2, 3, 4 and 6, result from behaviors of the northern limb only, while the features in segment 1 and 5 result from behaviors of both limbs. By further comparing with earlier works, we notice that the gradual recollimations in segment 1 between $d = 0.90$ and 1.30 mas and in segment 5 between $d = 3.65$ and 4.25 mas, spatially associate with recollimations in the first and second collimation regime (at $d = 1.1$ and 3.7 mas respectively) suggested by [8]. So they could possibly be the representations of these persistent recollimation features at our observed epoch.

### 3.2. Spectral-Index Distribution

In this part, we present the spectral-index distribution in the jet calculating between 22 and 43 GHz. To align the VLBI images at different frequencies, we employ an algorithm which is first introduced by [15]. It is based on the premise that the positions of optically thin regions to synchrotron radiation are not affected by absorption effects, then images can be aligned by finding out the maximum of 2D cross-correlation of the optically thin regions. This method is now commonly used [16–18] and was once employed by [19] to obtained the RM map of jet in M87.

The practical steps are listed as follow: (1) First, we produce VLBI images using the fully self-calibrated dataset. A same $u - v$ range is used to minimize the difference of baseline coverages between 22 and 43 GHz. Both images are restored with a circular beam with a diameter of 0.65 mas (the same as used in Section 3.1). (2) Second, we produce an initial spectral-index map by simply aligning the geometric center of two images. With the initial map, we roughly determined the optical thick region which will be masked in the next step. (3) Third, we import the two images into an interactive Python program, VIMAP [20], based on the algorithm introduced by [15]. A circular mask with a diameter of 1 mas is used to cover the core region, and a rectangle box with relative RA range from 2 to −4 mas and DEC range from −2 to 3 mas is used to choose a optically thin region. The VIMAP gives the values of 2D-correlation after shifting and find out the shifting corresponding to the maximum (see Figure 6). In our case, shifting corresponding to the maximum is zero. (4) The final spectral-index map is produced in AIPS with task COMB. Only regions with intensity $> 5\sigma$ are used to make sure the result is reliable and a noise map showing the distribution of errors is also produced. The 22–43 GHz spectral-index map superimposed on the intensity contour plots of 22 GHz are presented in panel (a) of Figure 7 with its noise map in the panel (b).

We also produce a 43–86 GHz spectral-index map with the same procedure. Considering there is a gap of six days between the observations of two frequencies, this spectral-index map is produced only for reference. In calculating the 2D cross-correlation, a circular mask with a diameter of 0.5 mas and a rectangle box with a RA range from 1 to −2 mas and a DEC range from −1 to 0.5 mas are used. The 43–86 GHz spectral-index map with its noise map are presented in the lower panel of Figure 7.

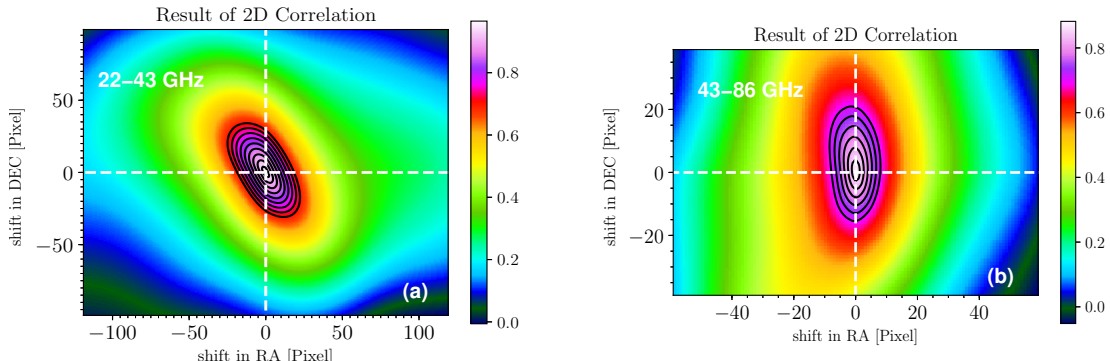

**Figure 6.** Panel (**a**) shows 2D cross-correlation of the optically thin regions of 22 and 43 GHz images. Panel (**b**) 2D cross-correlation of the optically thin regions of 43 and 86 GHz images.

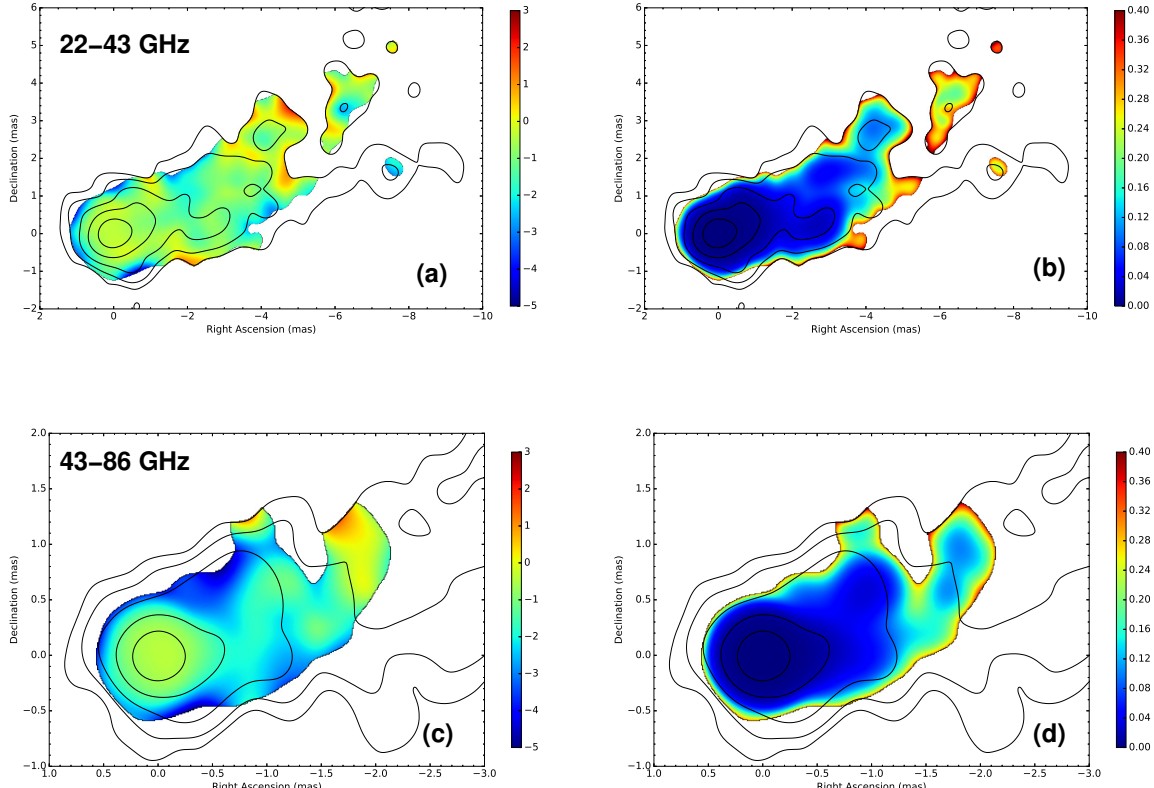

**Figure 7.** The pseudo-color maps of spectral-index distribution with color bars superimposed on the total intensity contour plots. Panel (**a**) shows 22–43 GHz spectral-index map superimposed on the contour plot of 22 GHz. Panel (**c**) shows the 43–86 GHz spectral-index map superimposed on the contour plot of 43 GHz. Their noise maps showing distribution of uncertainty are also presented in panel (**b**) and (**d**) respectively.

On the 22–43 GHz spectral-index map, inverted-spectrum with index $\alpha > 0$ ($S_{obs} \propto \nu^{\alpha}$) are detected between $d = 1.30$ and 1.60 mas on the edge of the northern limb, and between $d = 1.30$ and 1.80 mas on the edge of the southern limb. Some less confident inverted-spectrum are seen on the edge of northern limb around $d = 3.80$ mas. Strongly inverted spectrum with $\alpha > 2$ is detected in the northern limb between $d = 5.00$ and 7.00 mas where the limb is interrupted. These inverted-spectrum features indicate absorptions in the jet in M87. Interestingly, on the map, we see a pattern-like distribution along the jet, the regions with steep spectrum interlace with regions with flatter spectrum (the pattern-like distribution is also seen on the 43–86 GHz spectral-index map).

## 4. Discussion

### 4.1. Jet Recollimation with Absorption

As we have mentioned in Section 3.1, recollimations between $d = 0.90$ and 1.30 mas and between $d = 3.65$ and 4.25 mas could possibly be the representations of the persistent recollimations suggested by [8] at our observed epoch. By comparing our obtained transverse jet structure and the 22–43 GHz spectral-index map, we find these two recollimations are spatially close to the inverted-spectrum features.

The collimations are nonequilibrium behaviors in the AGN jet-launching region before the hydrodynamic or magnetohydrodynamic equilibrium is achieved, and they reminds us the reconfinement nodes in the simulations of a jet propagating in the external medium with pressure decreasing slower

than the thermal pressure of jet [21–28]. As to the external medium, since these features are between $300$–$2000R_s$ from the central engine (consider the viewing angle of the jet in M87 as $17°$, as used in [8]), in the hot accretion flow models like ADIOS (see [29] for an overview), the external medium interacting with an AGN jet on this scale is believed to be "winds", i.e., the moderately magnetized, non-relativistic un-collimated, extremely hot and generally fully ionized gas outflows launched from the accretion disk.

Since stationary shocks at the reconfinement node may enhance the magnetic fields locally [26–28], thus increase the opacity of Synchrotron Self-Absorption, SSA could be a possible mechanism responsible for the spectral turnover at the recollimations. But in a SSA-only case, according to Equation (2) of [30], for a region with intensity of 0.02 Jy Beam$^{-1}$ (a typical value of intensity of jet in M87 between $300$–$2000R_s$), turnover frequency higher than 43 GHz requires a unreasonably intense magnetic filed of $10^5$ $G$. So contribution of other mechanisms like free-free absorption from the external medium ("winds") must be substantial, therefore spectral turnover at the recollimations may also indicate a large free-free opacity, i.e., high density of external medium there, suggesting either the jet collimations locally enhance the density of the external medium, or the locally high density of external medium induces the collimations of the jet. (by looking back upon literatures, we find that the 22–43 GHz spectral-index map in [31] also showed an association of collimation and absorption at locations between $d = 1$ and 2 mas, but it was not reported explicitly then).

### 4.2. Jet Interruption with Absorption

As we have mentioned in Section 3, an interruption of the northern limb between 5 and 7 mas from the center of the image is seen at 22 GHz. On the restored 22 GHz image, the peak of the northern limb around $d = 6.0$ mas is below $5\sigma$. According to the results of 22 GHz KaVA monitoring and 15 GHz MOJAVE survey on M87, at many epochs in a period longer than 20 years, an interruption is seen in the northern limb somewhere between $d = 5$ and 10 mas, thus the jet in M87 shows a single-ridgeline morphology in this region, e.g., at 22 GHz, 2 March 2014 and 3 May 2014 [7]; at 15 GHz, 1 November 1998, 9 August 2004, 11 February 2010, 27 August 2019, et al.

Inverted spectrum with $\alpha$ up to 2.2 is detected around the interruption on our 22–43 GHz spectral-index map, suggesting a strong absorption feature in this area. Interestingly, we find that the inverted spectrum is also seen at a similar location around $d = 7$ mas on the 22–43 GHz spectral-index map in [31]. Since $\alpha$ is very close to the upper limit of SSA (if we set the cutoff to $3\sigma$, the $\alpha$ is almost 3), free-free absorption may probably be the main absorption mechanism here. Since this feature is between $2000$–$4000R_s$ from the central engine, "winds" are still believed to be the main absorption medium. With our intensity slices, we estimate the opacity of this absorption feature: on the restored 22 and 43 GHz images, the intensity peak of the northern limb at $d = 6.0$ mas is 1.5 mJy beam$^{-1}$ and 2.3 mJy beam$^{-1}$ respectively. If emissions were optically thin here, assuming the optically thin spectral index is $-2$, then the intensity peak in this region should be 8.8 mJy beam$^{-1}$ at 22 GHz. If $S_{obs} = S_{pre}e^{-\varnothing}$, then this implies an opacity of $\varnothing \sim 1.8$ at 22 GHz.

The interruption of the northern limb between 2000 and $4000R_s$ from the central engine is an indication that the external medium is not uniform in the jet-launching region. As the jet propagates outward, it may go through regions with high f-f opacity. In addition, the pattern-like spectral-index distribution along the jet might be an indication of non-uniform external medium as well.

### 4.3. Temporal Features and Short-Lived Instability

As we have mentioned in Section 3.1, on our 86 GHz image, the northern limb looks rather ragged. The analysis of transverse structure shows the northern limb starts at a larger core distance than its southern counterpart, and it has a tortuous and fragmentized structure, which means its orientation changes several

times abruptly, resulting in structural features in the jet segment 2, 3, 4 and 6. These phenomenon have never been reported by works based on long-term mm-VLBI monitoring, so they might just be temporal features, and represents of the variability of the AGN jet in its launching region.

It has been proved by mm-VLBI observations, that the flow in the jet in M87 is rotating clockwisely and with helical magnetic fields [8]. A local, internal kink instability can grow over short timescales in a jet like this and lead to oscillations [25,32–35]. At the observed epoch, a southward jet oscillation is found between $d = 0.75$ mas and 3.00 mas, and most clearly seen at $d = 1.45$ mas. So, the temporal features could possibly be attributed to the observed southward jet oscillation caused by the kink instability. The northern edge of the jet in M87 was compressed by the oscillation, thus local structure of the northern limb was damaged, and finally brought this fragmentized morphology of the northern limb and these temporal structural features at the observed epoch.

## 5. Conclusions

With the multi-frequency quasi-simultaneous VLBI observations on M87, for the fisrt time, we find a spatial association between the jet collimations and the local enhancement of the density of external medium ("winds") in the jet-launching region, suggesting either the jet collimations locally enhance the density of the external medium, or the locally high density of external medium induces the collimations of the jet. We also find that the external medium in the jet-launching region, is not uniform, and greatly contribute to the absorption in this region. In addition, we find that for a rotating jet with helical magnetic filed, like the one in M87, its temporal morphology in the launching region may be largely affected by the local, short-lived kink instability growing in itself.

**Author Contributions:** Conceptualization, X.H., W.Z., T.A., F.W.; data reduction and writing, W.Z.; software and visualization, W.Z., X.L.; doubel-check of data reduction, X.C.; funding acquisition, W.Z.

**Funding:** This research is funded by National Natural Science Foundation of China 11803062.

**Acknowledgments:** We sincerely thank the anonymous referee for her/his careful reviewing that improved the manuscript. The VLBA is an instrument of the National Radio Astronomy Observatory. The National Radio Astronomy Observatory is a facility of the National Science Foundation operated under cooperative agreement by Associated Universities, Inc. We also thank Kazuhiro Hada for his valuable suggestions on the data analysis. This work made use of the Swinburne University of Technology software correlator, developed as part of the Australian Major National Research Facilities Programme and operated under license.

**Conflicts of Interest:** The funders had no role in the design of the study; in the collection, analyses, or interpretation of data; in the writing of the manuscript, or in the decision to publish the results.

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
