# Peer review of "Features of Structure and Absorption in the Jet-Launching Region of M87"

_galaxies, doi:10.3390/galaxies7040086_

Round 1

Reviewer 1 Report

The authors present the results of 3-frequency VLBA observations of the jet of the galaxy M87. The loss of the MK station at 22 and 43 GHz limits the angular resolution of this study. Since the resolution at 22 and 43 GHz was higher in previous studies in the literature (cited by the authors), the paper should indicate explicitly, in both the introduction and conclusions, and possibly in the abstract, what the authors' observations contribute to the study of M87 beyond what is discussed in the papers by Asada and Nakamura, Hada et al., and Walker et al.

lines 100-118: An apparent core shift from resolution effects can occur if there are gradients in the intensity, but not if the intensity is uniform across the source. Of course, gradients are expected for a jet. This combination of gradients in intensity, coupled with angular resolution depending on frequency, should be mentioned as the cause of the apparent shift.

Author Response

Point 1
The authors present the results of 3-frequency VLBA observations of the jet of the galaxy M87. The loss of the MK station at 22 and 43 GHz limits the angular resolution of this study. Since the resolution at 22 and 43 GHz was higher in previous studies in the literature (cited by the authors), the paper should indicate explicitly, in both the introduction and conclusions, and possibly in the abstract, what the authors' observations contribute to the study of M87 beyond what is discussed in the papers by Asada and Nakamura, Hada et al., and Walker et al.

Response 1

The introduction, conclusion and abstract (and discussions as well) have been revised.

Our contributions (like "for the first time, we compare the results of the two analyses and find a spatial association between the jet collimations and the local enhancement of the density of external medium in the jet-launching region") are addressed more explicitly in the revised version.

The revisions are in italic  type

Point 2

lines 100-118: An apparent core shift from resolution effects can occur if there are gradients in the intensity, but not if the intensity is uniform across the source. Of course, gradients are expected for a jet. This combination of gradients in intensity, coupled with angular resolution depending on frequency, should be mentioned as the cause of the apparent shift.

Response 2

It has been revised as your suggestions.

The revisions are in italic  type

ps: the revisions in bold type is for another referee

Reviewer 2 Report

Comments to Authors:

In the manuscript, "Features of Structure and Absorption in the Jet-launching Region of M87" by Zhao et al., authors have used their new observations of M87 taken in 2015 and the finding of the manuscript is interesting. The manuscript may be considered for publication after following minor revision in the draft.

Line 1: M87 is the best ---> M87 is one of the best

Line 12: Here authors have reported the upper limit of the BH mass of AGN reported yet. Since M87 is a very well studied AGN, instead of upper limit of BH mass, detected BH mass range should be reported. The most accurate BH mass estimation in AGN can be done by using reverberation mapping (RM), authors must search the published literature and quote the value of BH mass using RM if it is done. Other BH mass estimation can be done by using stellar gas dynamics, and scaling relation, authors must report the BH masses by these methods too.

Line 16: Hada et al.[3] ---> [3]

Line 17: by EHT Collaboration 2019 [4] ---> [4]

Line 20: the variable details ---> the variablity details

Line 29: Very Long VLBI Array (VLBA) ---> very long baseline array (VLBA)

Line 29: almost simultaneously ---> quasi simultaneously

Table 1 and column no. 3: Instead of using -MK, -HN, -NL, -SC here, please use that in the footnote of table and also explain there.

Line 37-40: Mauna Kea was absent due to the "bad diskpack" problem, Hancock and Saint Croix have no 86 GHz receiver while NL was "disc failure, must be given in footnote of Table 1 as I suggested in previous comment. So, no need to write here.

Figure 1 and 2: The pseudo color image of M87 on which contours are plotted is taken in which electromagnetic band, please mention in the figure caption as well as in text.

Line 73: as in Lee et al. [11] ---> as in [11]

Line 74: Güijosa & Daly [12] ---> [12]

Line 76: Hada et al. [8] and Kim et al. [7] ---> [7,8]

Line 120: Asada & Nakamura [9] and Hada et al [10] ---> [9,10]

Line 142: Walker et al. [6] ---> [6]

Line 146-147: which is first introduced by Croke & Gabuzda [13] --> first introduced in [13]

Line 150: Park et al.[17] ---> [17]

Line 156: by Croke & Gabuzda [13] ---> in [13]

In general there is no need to authors name and then reference number. Please only cite reference number.

Authors must cite the references starting from lower to higher numbers. It seems references are taken in arbitrary sequence.

Author Response

Point 1
Line 1: M87 is the best ---> M87 is one of the best

Response 1

It has been revised as your suggestion, and the revision is in bold type.

Point 2
Line 12: Here authors have reported the upper limit of the BH mass of AGN reported yet. Since M87 is a very well studied AGN, instead of upper limit of BH mass, detected BH mass range should be reported. The most accurate BH mass estimation in AGN can be done by using reverberation mapping (RM), authors must search the published literature and quote the value of BH mass using RM if it is done. Other BH mass estimation can be done by using stellar gas dynamics, and scaling relation, authors must report the BH masses by these methods too.

Response 2

Sorry I can not find out the BH mass derived using RM, so I just add two references here (Harms et al. 1994 and Gebhardt et al. 2011), in which the BH mass is derived with gas and stellar dynamics respectively.
And the revision is in bold type.

Point 3
Line 16: Hada et al.[3] ---> [3]

Response 3

It has been revised as your suggestion.

Point 4
Line 17: by EHT Collaboration 2019 [4] ---> [4]

Response 4

It has been revised as your suggestion.

Point 5
Line 20: the variable details ---> the variablity details

Response 5

It has been revised as your suggestion, and the revision is in bold type.

Point 6
Line 29: Very Long VLBI Array (VLBA) ---> very long baseline array (VLBA)

Response 6

It has been revised as your suggestion, and the revision is in bold type.

Point 7
Line 29: almost simultaneously ---> quasi simultaneously

Response 7
It has been revised as your suggestion, and the revision is in bold type.

Point 8
Table 1 and column no. 3: Instead of using -MK, -HN, -NL, -SC here, please use that in the footnote of table and also explain there.

Response 8

It has been revised as your suggestion.

Point 9
Line 37-40: Mauna Kea was absent due to the "bad diskpack" problem, Hancock and Saint Croix have no 86 GHz receiver while NL was "disc failure, must be given in footnote of Table 1 as I suggested in previous comment. So, no need to write here.

Response 9

It has been revised as your suggestion. Two sentences are added here to make it read fluently. "For all three frequencies, some antennas were absent during the observations due to some technical problems. The details of the problems are described in the footnote of Table 1"

The added sentences are in bold type.

Point 10
Figure 1 and 2: The pseudo color image of M87 on which contours are plotted is taken in which electromagnetic band, please mention in the figure caption as well as in text.

Response 10

It has been revised as your suggestion, and the revision is in bold type.

Point 11
Line 73: as in Lee et al. [11] ---> as in [11]

Response 11

It has been revised as your suggestion.

Point 12
Line 74: Güijosa & Daly [12] ---> [12]

Response 12

It has been revised as your suggestion.

Point 13
Line 76: Hada et al. [8] and Kim et al. [7] ---> [7,8]

Response 13

It has been revised as your suggestion.

Point 14
Line 120: Asada & Nakamura [9] and Hada et al [10] ---> [9,10]

Response 14

It has been revised as your suggestion.

Point 15
Line 142: Walker et al. [6] ---> [6]

Response 15

It has been revised as your suggestion.

Point 16
Line 146-147: which is first introduced by Croke & Gabuzda [13] --> first introduced in [13]

Response 16

It has been revised as your suggestion.

Point 17
Line 150: Park et al.[17] ---> [17]

Response 17

It has been revised as your suggestion.

Point 18
Line 156: by Croke & Gabuzda [13] ---> in [13]

Response 18

It has been revised as your suggestion.

Point 19
In general there is no need to authors name and then reference number. Please only cite reference number.

Response 19

It has been revised as your suggestion.

Point 20
Authors must cite the references starting from lower to higher numbers. It seems references are taken in arbitrary sequence.

Response 20

It has been revised as your suggestion.

ps: the revisions in italic type is for another referee